# Assessment of the environmental impacts of conflict-driven Internally Displaced Persons: A sentinel-2 satellite based analysis of land use/cover changes in the Kas locality, Darfur, Sudan

Abdalrahman Ahmed[1,2]*, Brian Rotich[3,4], Kornel Czimber[1]

1 Faculty of Forestry, Institute of Geomatics and Civil Engineering, University of Sopron, Sopron, Hungary,
2 Department of Forest and Environment, Faculty of Forest Science and Technology, University of Gezira, Wad Madani, Sudan, 3 Institute of Environmental Sciences, Hungarian University of Agriculture and Life Science, Gödöllő, Hungary, 4 Faculty of Environmental Studies and Resources Development, Chuka University, Chuka, Kenya

* ix3mzk@uni-sopron.hu

**Data Availability Statement:** All relevant data are within the manuscript and its Supporting Information files.

## Abstract

Internal displacement of populations due to armed conflicts can substantially impact a region's Land Use and Land Cover (LULC) and the efforts towards the achievement of Sustainable Development Goals (SDGs). The objective of this study was to determine the effects of conflict-driven Internally Displaced Persons (IDPs) on vegetation cover and environmental sustainability in the Kas locality of Darfur, Sudan. Supervised classification and change analysis were performed on Sentinel-2 satellite images for the years 2016 and 2022 using QGIS software. The Sentinel-2 Level 2A data were analysed using the Random Forest (RF) Machine Learning (ML) classifier. Five land cover types were successfully classified (agricultural land, vegetation cover, built-up area, sand, and bareland) with overall accuracies of more than 86% and Kappa coefficients greater than 0.74. The results revealed a 35.33% (-10.20 km$^2$) decline in vegetation cover area over the six-year study period, equivalent to an average annual loss rate of -5.89% (-1.70 km$^2$) of vegetation cover. In contrast, agricultural land and built-up areas increased by 17.53% (98.12 km$^2$) and 60.53% (5.29 km$^2$) respectively between the two study years. The trends of the changes among different LULC classes suggest potential influences of human activities especially the IDPs, natural processes, and a combination of both in the study area. This study highlights the impacts of IDPs on natural resources and land cover patterns in a conflict-affected region. It also offers pertinent data that can support decision-makers in restoring the affected areas and preventing further environmental degradation for sustainability.

## Introduction

Land use and land cover (LULC) changes are increasingly being viewed as the primary driver of global environmental changes, including greenhouse gas emissions, global climate change,

**Funding:** The author(s) received no specific funding for this work.

**Competing interests:** The authors have declared that no competing interests exist.

biodiversity, and ecosystem services loss, soil resource loss, and livelihood loss [1–5]. LULC changes are caused by either natural forces, anthropogenic drivers, or both. Changes in LULC occur primarily due to human pressures on natural landscapes at various geographical and temporal scales as influenced by multiple factors that differ by region [6–8]. Natural causes of LULC changes comprise droughts, natural fires, and general changes and variations in climatic conditions while the common anthropogenic drivers of LULC changes include agricultural expansion, deforestation, urbanization, mining, and armed conflicts [9–12].

Armed conflicts are an extreme form of socioeconomic shocks that can shape future land-use trajectories [13]. These conflicts can have substantial effects on LULC, especially in the conflict zones, and in areas where the displaced population settles [13]. Globally, armed conflicts have resulted in severe demographic, social, economic, and political implications on the lives and livelihoods of the affected populations. At the landscape level, armed conflicts can either trigger increased or reduced pressure on land-based resources, especially vegetation and wildlife [12, 14, 15]. Warfare can benefit natural systems due to reduced human pressure on the natural environment in the conflict region thereby reducing contamination and environmental degradation [12, 16]. On the other hand, the induction of displaced populations from conflict zones into new landscapes can result in the over exploitation of natural resources leading to deforestation, water pollution, agricultural expansion, and land fragmentation [17–20].

Monitoring and mapping changes in LULC is pivotal for environmental management [12]. Satellite Remote Sensing (RS) data integrated with Geographical Information System (GIS) is an important tool for mapping the extent and spatial distribution of forest and land uses based on a stable classification system for change and analysis with the integration of field data [21, 22]. It is an adequate tool for monitoring and detecting changes in vegetation cover using multi-temporal data. Images from previous years can be compared to recent years to measure the differences in the sizes and extent of forest cover [21, 23]. The use of RS technology is therefore considered a suitable approach for assessing historical and future LULC changes [23]. The advancement in satellite RS technology has revolutionized the approaches to monitoring the Earth's surface [24]. Since its launch in 2015, there has been a high adoption and application of Sentinel-2 images in LULC studies. This can be attributed to its free access policy, high spatial resolution (10 m), and the availability of red-edge bands with multiple applications [24, 25]. Sentinel-2 data can also integrate with other remotely sensed data, as part of data analysis, which improves the overall accuracy when working with Sentinel-2 images. In addition, when used with machine-learning classifiers such as Random Forest (RF) and support vector machine (SVM), Sentinel-2 data produces high accuracies (>80%) [24].

The semi-arid Darfur region of Sudan has experienced a protracted humanitarian crisis, characterized by armed conflict, widespread displacement, and environmental degradation since the early 2000s [26, 27]. A new wave of violence and insecurity emanating from fighting involving armed movements, government forces, and armed tribal militia has rocked Darfur since 2014. This conflict has created an additional humanitarian crisis in the region as it has led to the displacement of about 322,000 people [27]. The conflict in Darfur has greatly accelerated the processes of environmental degradation that have undermined subsistence livelihoods and the environmental conditions in the area over the recent decades. The emergence of numerous internally displaced persons (IDPs) communities around the major towns of the Darfur region during the conflict has led to the unsustainable utilization of natural resources significantly impacting the environment through agricultural expansion, overgrazing, and deforestation leading to environmental degradation [28, 29]. This environmental degradation has not only exacerbated the vulnerability of IDPs but also contributed to broader ecological and climatic concerns in the region thereby curtailing the efforts in achieving the Sustainable Development Goals (SDGs) [26, 30]. Despite these impacts, little research has been done to

quantify and map the recent vegetation cover changes in the region because of IDPs settlement since the conflict resurgence in the year 2014.

Previous vegetation cover change research in Darfur [29, 31, 32] was done before 2016. All those studies were conducted in Central, Western, and North Darfur. Our study is the first in South Darfur after the launch of Sentinel-2 satellite data for public use This study therefore aims to provide the latest essential post 2014 spatial data on LULC changes in the Kas locality of Darfur, which is lacking owing to the ongoing conflicts. The results will further enhance the understanding of the repercussions of the 2014 conflict-driven IDPs on vegetation cover and overall environmental sustainability in the Kas locality of Darfur, Sudan. The findings of this study are crucial for informing decision-makers and guiding efforts to restore the affected areas, mitigate further environmental degradation, and contribute to the broader objective of achieving (SDGs) in conflict-affected regions.

## Methods

### Study area

The study area is located between latitudes 12˚23´ and 12˚36´north and longitudes 24˚9´and 24˚26´east, covering an estimated area of about 1007.13 km$^2$ (Fig 1). It lies at an altitude of 400 meters above sea level [33]. It is situated in Kas, South Darfur state, around 86 kilometers northwest of the state's capital Nyala [34]. The Kas location has an its average temperature of 26˚C and average annual rainfall is 519 mm, which varies from north to south (Fig 2). The total population of the locality is estimated at 365,000 people, 76,843 of whom are IDPs [35]. The two major natural resource-based livelihood systems in Darfur are rain-fed agriculture and pastoralism [36]. The Acacia species dominate the forested areas, for instance the Kas Forest reserve. Darfur's labor market continues to be heavily reliant on resources from the forest and rangelands, placing an undue strain on such resources [36].

### Data collection and analysis

Sentinel-2 satellite images with a spatial resolution of 10 meters B2 (Blue), B3 (Green), B4 (Red), and B8 (NIR) were used in this study. These images were downloaded from the United States Geological Survey (USGS) website (http://earthexplorer.usgs.gov/) at Path 178 and Row 52 (Table 1). The dates of the images were chosen to be in the same month and the cloud cover was less than 1%. Rainfall and temperature data were sourced from power data access from the National Aeronautics and Space Administration (NASA) website (https://power.larc.nasa.gov/data-access-viewer/).

**Image processing.** Sentinel-2 Satellite images were selected for this study because of their high resolution (10 meters). The two downloaded satellite data were Level-2A products, and the available Sentinel-2 Satellite images of 8$^{th}$ of December 2016 and 2022 were used in this study (Fig 3). Image processing techniques included atmospheric correction, image classification, change detection, and accuracy assessment of the classified images were successfully achieved. All processing steps were carried out in QGIS 3.22 software. Microsoft Office Excel 2013 was used to compute the land cover changes to show the percentages and change rates.

**Machine learning algorithms.** A total of three classifiers—random forest (RF), support vector machine (SVM), and k-nearest neighbor (KNN)—were used to classify the Sentinel-2 images and their classification outputs compared. The RF classifier is an ensemble method using decision trees as classifiers. RF uses a large number of decision trees each is feature-aggregated (bagging) by bootstrapping the training samples. The final output is determined by the majority of the trees. [38, 39]. The Dzetsaka classification tool in QGIS was utilized in this study to apply RF classification to the Sentinel-2A images. The RF is a powerful learning

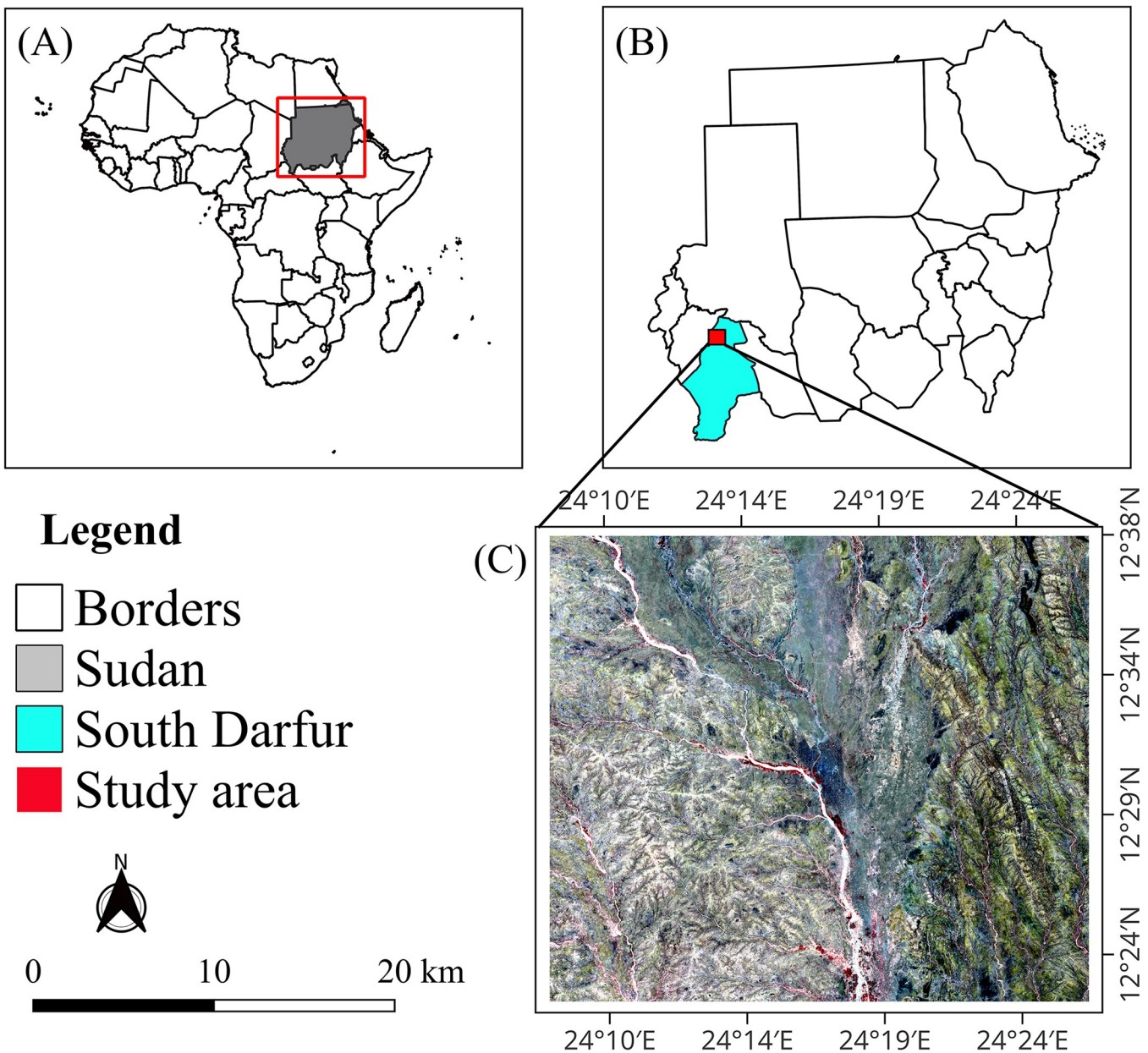

**Fig 1.** Map of the study area showing (A) Location of Sudan in Africa (B) Location of South Darfur in Sudan (C) Sentinel-2 Image of the study area.

technique since it applies feature significance properties and averages several predictions. A fixed number of 100 trees is set by default, and this has shown to be an appropriate size to avoid overfitting. [40–42]. During the construction each tree is split at every internal node using the square root of the number of features (n):

$$\text{max\_features} = \sqrt{(n\_features)} \tag{1}$$

From the training data (n_features), which consists of the pixel values defined by the ROIs (regions of interest), the features are determined and chosen at random. Thus, a suitable size

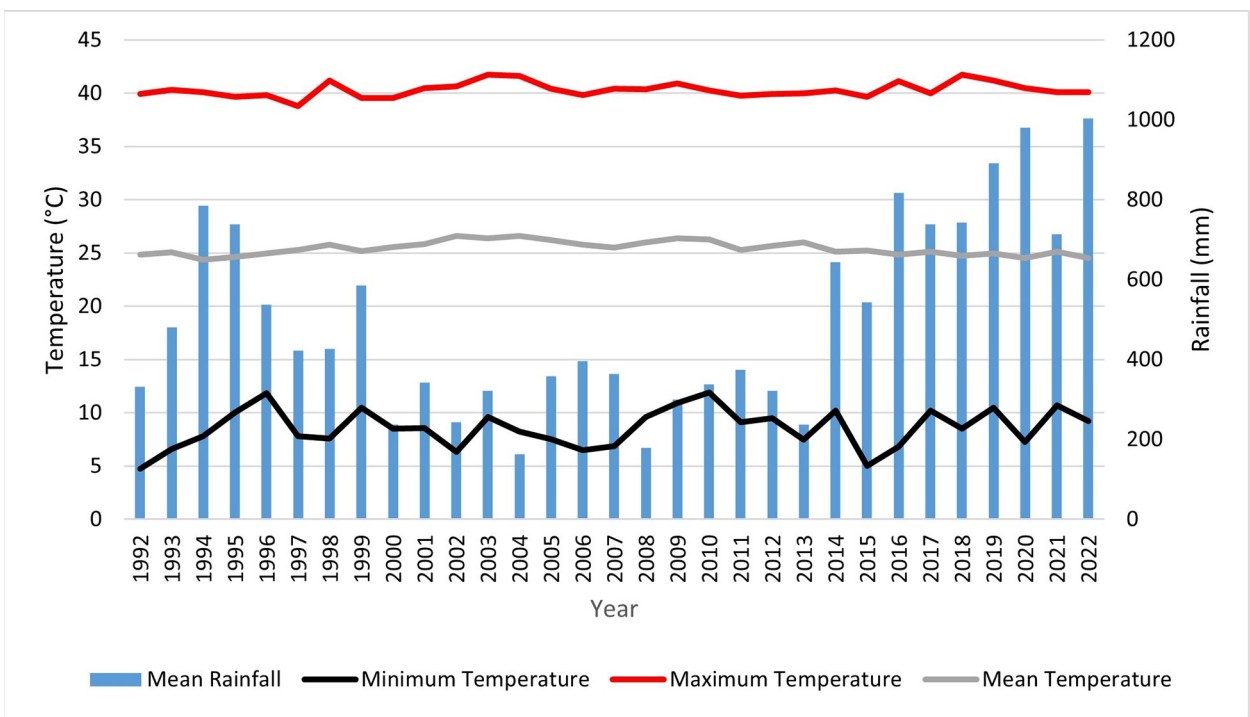

**Fig 2. Mean annual rainfall and temperature time series data of the study area from 1992–2022 [37].**

for the training data set should be ensured. There is no predetermined size for this; it relies on the specific attributes of each data set and the intended class number. [43–45].

The KNN is a nonparametric memory-based supervised machine learning classifier. Following the calculation of the number of neighbors for which k is an integer value [46] KNN is used to solve both classification and regression problems [47]. This parameter (number of neighbors) in the Dzetsaka plugin is selected using a cross-validation technique to optimize the quality of output. [48]. KNN classification was applied to the study area using QGIS's dzetsaka classification tool.

SVM, is a linear model for classification and regression problems that is mainly based on kernels, it was developed by Cortes and Vapnik [49]. The Gaussian kernel known as the radial basis function is the one used in Dzetsaka and provides high quality results for classifying tree species. [48]. SVMs have been utilized in this study as it widely used in remote sensing [50]. A mathematical formulation of the SVM can be found in Scikit learn [44, 46].

**Image classification and analysis.** Supervised classification was performed on Sentinel-2 satellite images acquired for the years 2016 and 2022. The supervised classification was performed using the Dzetsaka plugin in QGIS. The Dzetsaka classification plugin allows the user to classify images with several machine learning algorithms. However, to use all the algorithms in Dzetsaka, some dependencies had to be installed. The Scikit-learn 1.0.1 Python package is the reference Python library for machine learning. [44, 45]. Three classifiers RF, KNN, and

**Table 1. Details of Sentinel-2 satellite image used in this study.**

| Sensor | Acquisition Date | Path/Row | Grid cell size (Meters) | Bands |
|---|---|---|---|---|
| **Multi-spectral instrument (MSI)** | 12/02/2016 | 178/52 | 10 | 2,3,4,8 |
| **Multi-spectral instrument (MSI)** | 20/02/2022 | 178/52 | 10 | 2,3,4,8 |

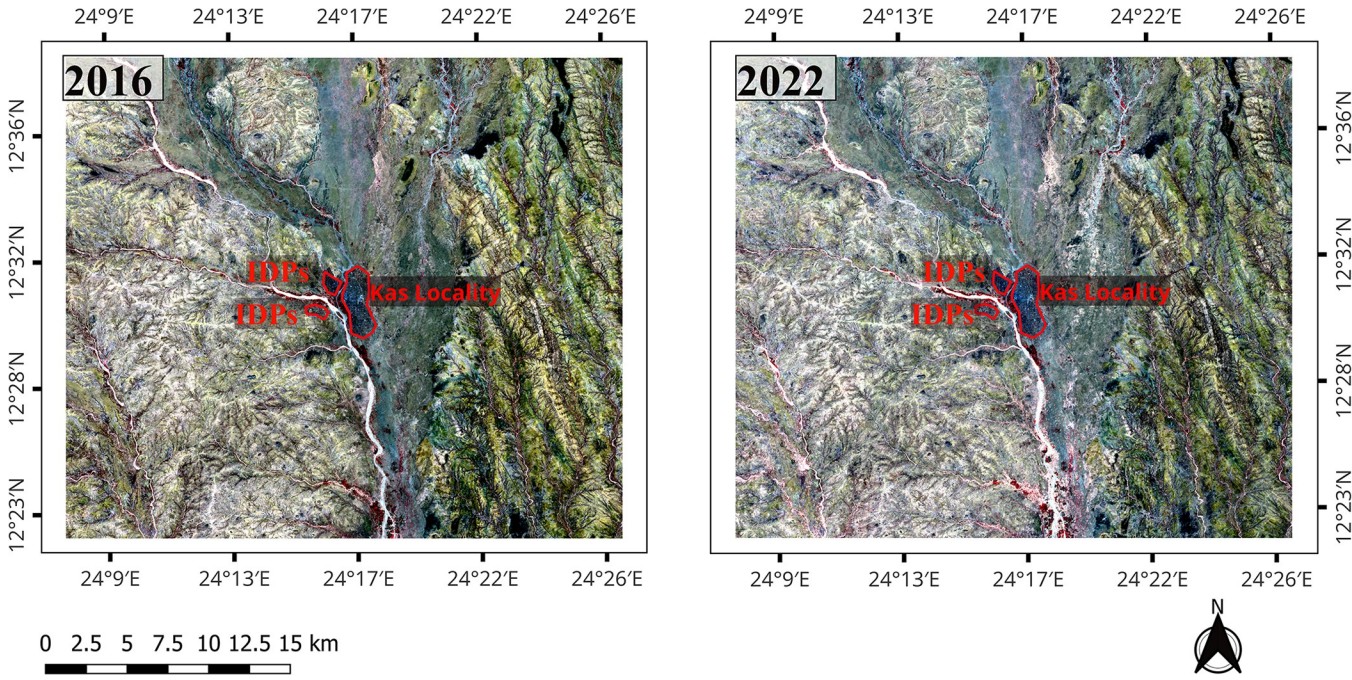

**Fig 3. Sentinel 2 satellite bands combinations (Red: B8, Green: B4, and Blue: B3) for the year 2016 and 2022.**

SVM were run on both Sentinel 2 images, and the output raster was generated. Default parameters of Dzetsaka plugin were used to run all algorithms [43, 46]. Before image classification we used raster calculator tool in QGIS to calculate the NDVI (Normalised Different Vegetation Index) using Eq (2) [51]. The NDVI index is an index used for evaluation of vegetation health as it shows biomass production. Note that in Sentinel-2 Scene the Red is band 4, the Near Infra-Red (NIR) is band 8.

$$NDVI = \frac{NIR - Red}{NIR + Red} \tag{2}$$

The NDVI values range between -1 and +1 where positive values represent healthy vegetation, values close to zero are bareland, and negative values represent water and clouds. Observed values from our analysis ranged from 0 to 0.6 (Fig 4).

Manual digitization of vegetation cover, built-up, sand, agricultural land and bareland areas was performed based on high spatial resolution satellite imagery available through the Google Earth Pro software using the historical imagery slider to move between the acquisition dates 2016 and 2022, additionally we used different Sentinel-2 Bands colour composite. The NDVI was assessed and used to increase classification accuracy. The NDVI threshold for the separation of vegetation and the historical high-resolution images on Google Earth platform were used as a reference data [41, 52] The onscreen digitization approach has been widely used and reported in previous studies for obtaining LULC classes, and it has been found to be reliable and accurate [53, 54]. We generated 202 samples for the year 2016 and 216 samples for the year 2022, representing the five LULC classes. Then we extracted the corresponding input pixels for both training and testing samples. The training samples were determined using a user defined number of pixels based on the proportional of the polygons per classes. The images were successfully classified into five classes namely, vegetation cover, bareland, built-up, agricultural land and sand (Table 2, Fig 5). The Sentinel 2 satellite data used in this study were

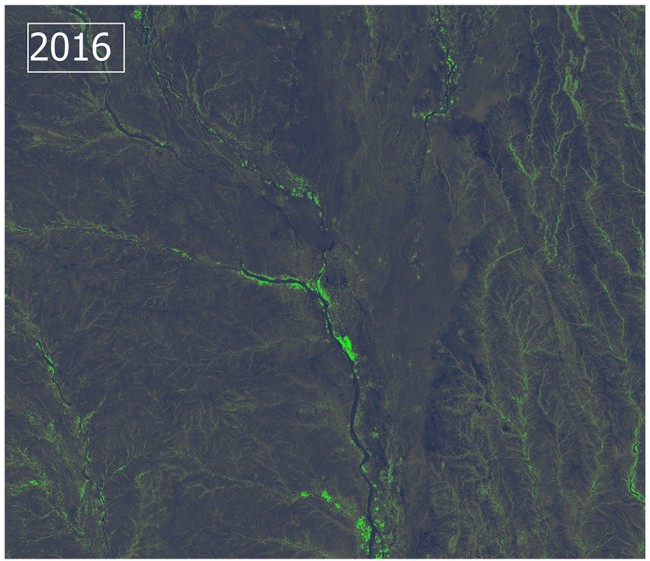
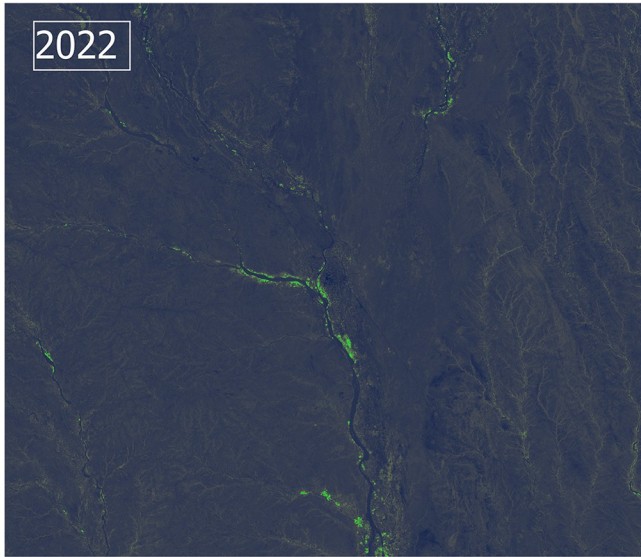

NDVI 2016 - 2022

0.6  High

0  Low

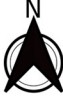
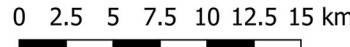

**Fig 4. NDVI map.**

analysed using QGIS version 3.28.2, software in combination with Microsoft Office Excel 2016, which was used in computing the LULC changes to show the percentages and change rates in the study area. A classification report containing the area of each class was then generated from the Semi-automatic Classification Plugin (SCP) in QGIS.

After the LULC was successfully classified, we subsequently used the Majority/minority filter in QGIS's SAGA tool to isolate the built-up class using a 20 pixels radius. A raster buffer was made 10 km from the center of the IDPs settlements (Fig 6) for comparative analysis of the vegetation cover in the immediate 10 km and next 5 km buffer zones and to establish which of the two zones was highly affected. This analysis was conducted to help us better

**Table 2. Description of LULC classes.**

| LULC Class | Description |
|---|---|
| **Vegetation cover** | Land covered by forests and shrubs. Forests comprise areas >1.5ha with more than 10% of tree cover, with a height exceeding 2 meters. Common tree species include *Acacia spp*, *Balanites aegyptiaca*, and *Eucalyptus spp*. Shrubs are small trees, with heights less than 2 meters, mainly acacia species such as *acacia mellifera*. |
| **Agricultural land** | Land with less than 10% of scattered vegetated cover, usually cultivated for crop production by the IDPs and the host community during the rainy season (June-October). |
| **Built-up** | Areas covered by temporary structures, semi-permanent buildings, and permanent buildings. Tents and sheds are examples of temporary structures while semi-permanent buildings comprise huts established for IDPs settlement. Permanent buildings are mainly occupied by the host community. |
| **Bareland** | Areas without vegetation covered by exposed soils, rocks, rough roads, or degraded lands |
| **Sand** | Lands dominated by sand with no vegetation cover and no agricultural activities. It includes seasonal water streams (wadies). |

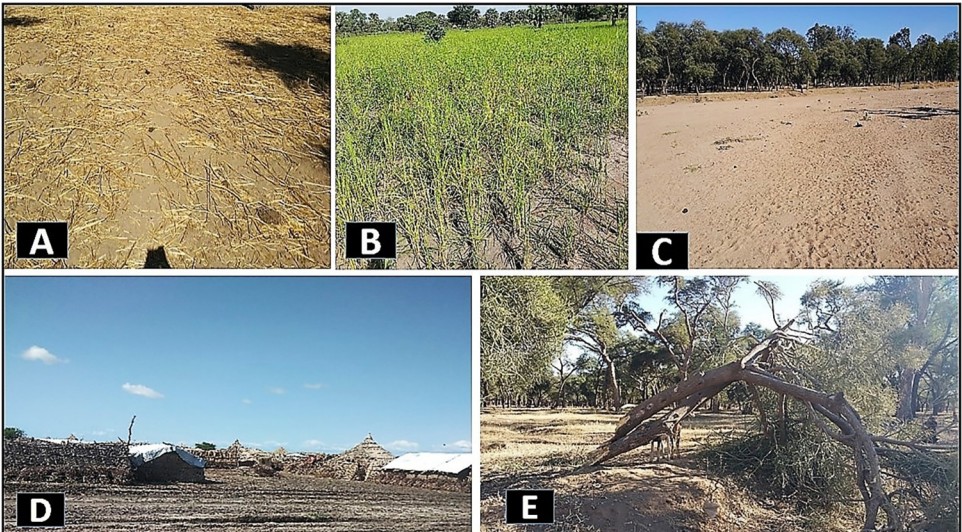

**Fig 5.** Photos showing the different land cover classes; (A) Bareland (B) Agricultural land (C) Sand with vegetation in the background (D) Built-up (E) Vegetation Cover.

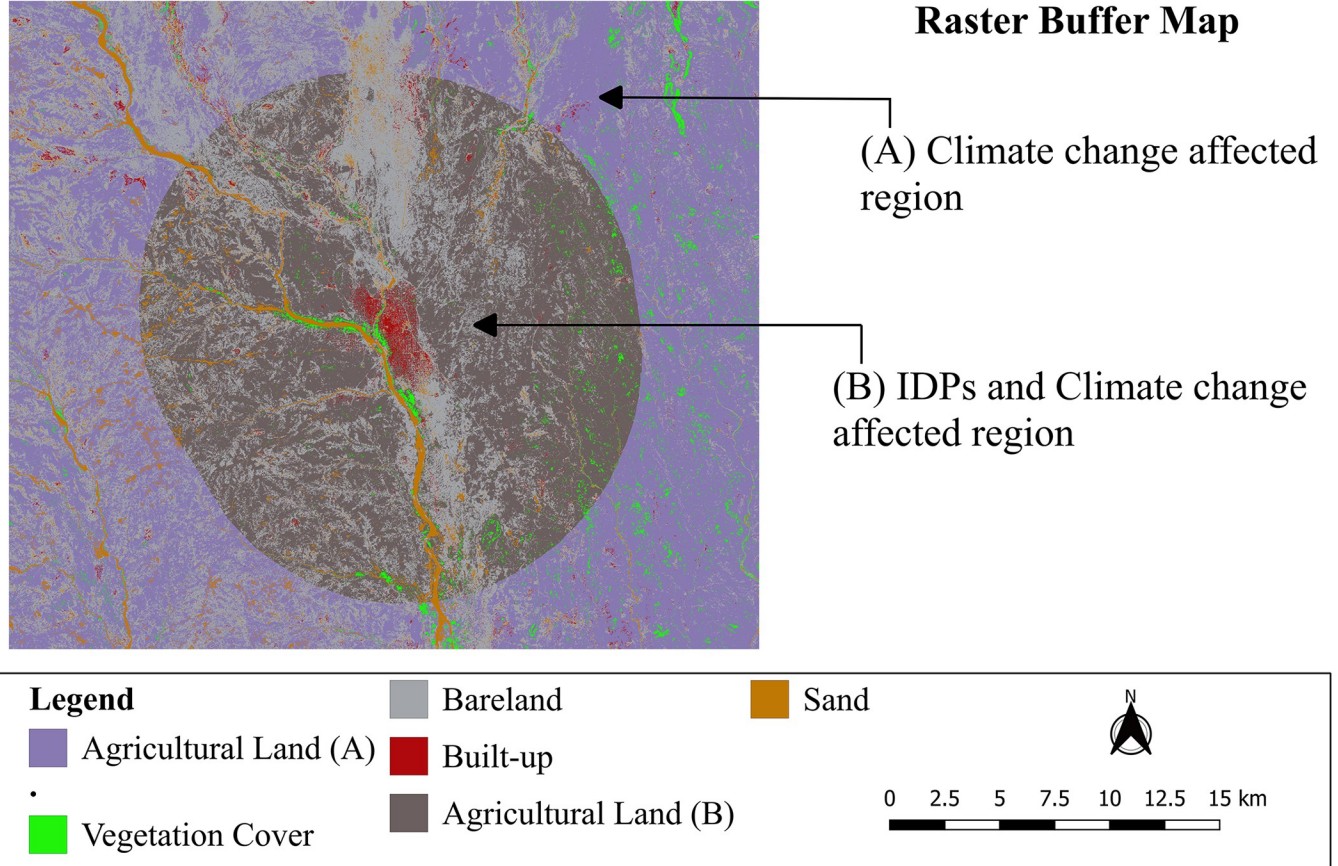

**Fig 6. Raster buffer map, showing the climate change affected region represent lowly affected region and IDPs with climate change affected region represent highly affected region.**

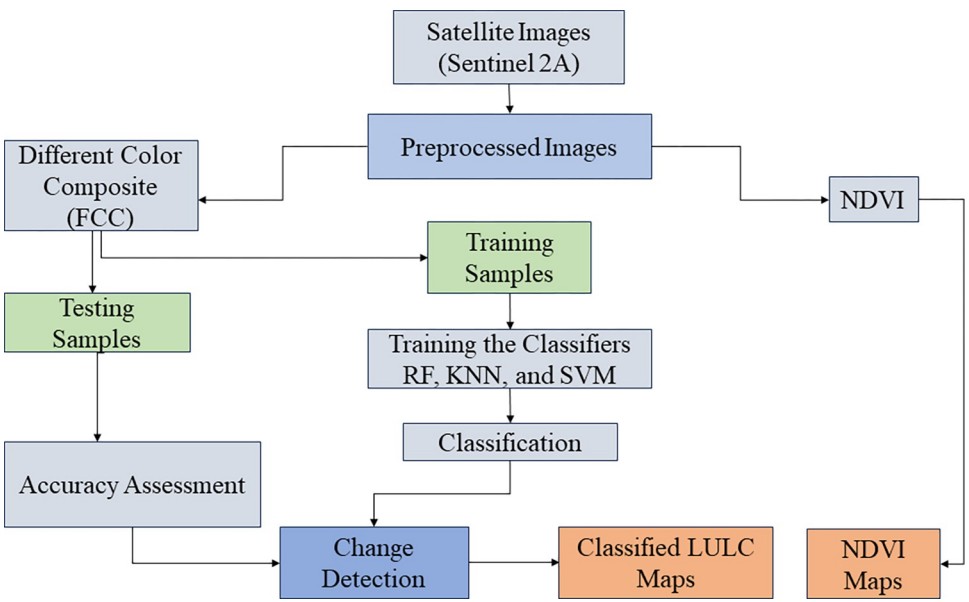

**Fig 7. Workflow used to analyze land use and land cover changes.**

understand the changes in vegetation cover caused by the IDPs as opposed to other underlying drivers.

**Accuracy assessment.** Accuracy assessment is one of the most important steps in the classification process. The aim of accuracy assessment is to quantitatively assess how effectively the pixels were sampled into the correct land cover classes. A total of 5092 pixels were collected for 2016, and 5344 pixels for 2022, then random stratified sample of an equal number of pixels (50%-50%) was applied in each land cover class to train the classifier, and to test it by calculating the confusion matrix. Confusion matrix is the most common way to present the accuracy of classified images [55]. Overall, users' and producers' accuracies, and Kappa values were generated from the error matrices. The user's accuracy is calculated by dividing the total number of classified points that agree with the reference data by the total number of classified points for that class. The producer's accuracy is calculated by dividing the total number of classified points that agree with reference data by the total number of reference points for that class. The Kappa value incorporates the off-diagonal elements of the error matrices and represents agreement acquired after removing the proportion of agreement that could be expected to occur by chance. Kappa statistic of the agreement provides an overall assessment of the accuracy of the classification. The Kappa coefficient can be negative and range from 0 (showing no agreement) to 1 (perfect agreement). The kappa value should be around zero for a fully random classification, and negative for a worse classification (than a random classification) [56, 57]. The data processing workflow for this study is summarized in Fig 7.

## Results

### LULC classification accuracy

Table 3 displays the accuracy of the various classes based on the different classifiers. It was discovered that the range of overall accuracy using various algorithms was 74% to 87% (Table 3). The RF algorithm, with an overall accuracy of 86.31% and a kappa of 0.75, demonstrated the best classification performance among the three classifiers in the 2016 LULC classes. Similarly, with a kappa of 0.71 and an overall accuracy of 85.52%, RF also mapped the 2022 classes. The

**Table 3. Overall accuracy (%) and kappa coefficient values of three different Machine Learning (ML) algorithms used to classify Sentinel 2 satellite imagery.**

| | 2016 | | 2022 | |
|---|---|---|---|---|
| **ML Algorithm** | **Overall Accuracy (%)** | **Kappa coefficient** | **Overall Accuracy (%)** | **Kappa coefficient** |
| **Random Forest (RF)** | 86.31 | 0.75 | 85.52 | 0.71 |
| **K-Nearest neighbor (KNN)** | 82.26 | 0.68 | 85.09 | 0.69 |
| **Support vector machine (SVM)** | 74.31 | 0.56 | 87.34 | 0.74 |

KNN algorithm had the lowest accuracy in 2016 and 2022, with 82.26% and 85.09% with kappa 0.68 and kappa 0.69, respectively. The RF results were the best among the three classifiers and were within the acceptable range hence we proceeded and used the classification outputs from the RF for this study [56].

## Land use land cover classification and spatial distribution

The results of Sentinel-2 satellite image classification showed that the total coverage of the study area was 1007.13 km$^2$. Individual class areas and percentages for the years 2016 and 2022 are summarized in Table 4. In 2016, agricultural land was the dominant LULC class covering slightly above half (55.56%) of the study area while bareland was the second largest land cover type at 34.97%. Vegetation cover and built-up areas had the least area coverage at 2.87% and 0.87% respectively. Similarly in 2022, the greatest share of LULC was agricultural land (65.32%) followed by bareland (28.33%), while built up areas had the least cover (1.39%).

The five LULC classes in Kas's locality for the years 2016 and 2022 were also mapped, and their spatial distribution patterns displayed in Fig 8. Built-up areas are located at the center of the study area while vegetation is mainly found in the eastern part of the Kas locality. Agricultural land dominates the eastern, sand the western and bareland the northern parts of the study area (Fig 8).

## Land use land cover changes

The magnitude and rate of changes in each LULC class between 2016 and 2022 are presented in Table 5. There was an overall reduction in the area under vegetation cover (-10.20 km$^2$), bareland (-66.85 km$^2$), and sand (-26.37 km$^2$). Conversely, agricultural land and built-up areas increased by 98.12 km$^2$ and 5.29 km$^2$ respectively. Vegetation cover had the second highest mean annual decline rate of -5.89% which translates to a loss of 1.70 km$^2$ of vegetation cover per annum. Agricultural land gained 16.35 km$^2$ (2.92%) per year.

The spatial distribution of vegetation cover changes to other classes and vice versa between 2016 and 2022 is visualized on Fig 9

**Table 4. LULC classes areas and percentages.**

| LULC class | 2016 | | 2022 | |
|---|---|---|---|---|
| | **Area (km$^2$)** | **Percent (%)** | **Area (km$^2$)** | **Percent (%)** |
| **Vegetation Cover** | 28.87 | 2.87 | 18.67 | 1.85 |
| **Bareland** | 352.21 | 34.97 | 285.36 | 28.33 |
| **Built-up** | 8.74 | 0.87 | 14.03 | 1.39 |
| **Agricultural Land** | 559.63 | 55.56 | 657.75 | 65.32 |
| **Sand** | 57.67 | 5.73 | 31.30 | 3.11 |
| **Total** | 1007.13 | 100.00 | 1007.13 | 100.00 |

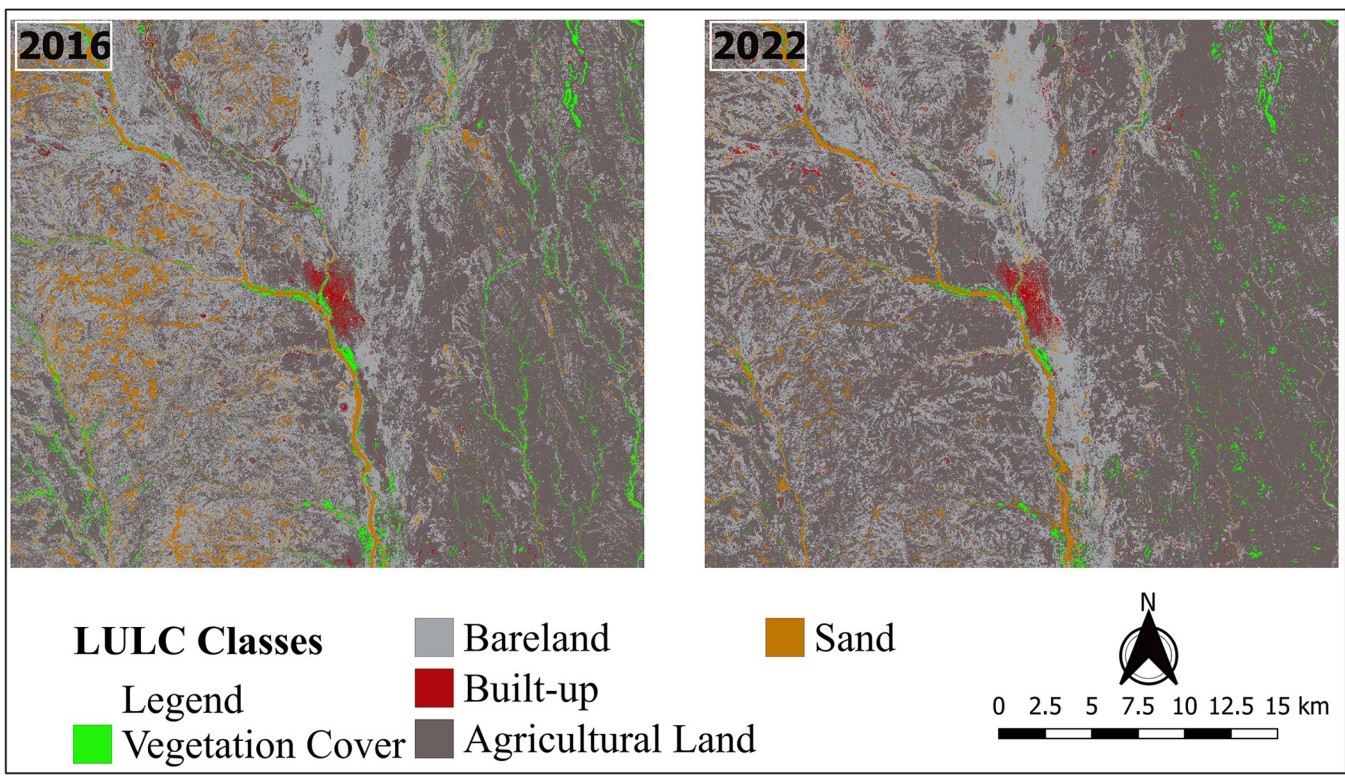

**Fig 8. The classified maps of Kas locality for 2016 and 2022.**

## IDPs and climate change influences on LULC

Results from the two buffer zones analysis (Table 6) show the percentage area and actual area (in square kilometers) for the different land cover types in both IDPs and climate change affected areas. It also shows the highly and lowly affected areas, along with the change percentages over the specified time frame. In the highly affected area (0–10 km), there was a -42.57% decrease in vegetation cover from 2016 to 2022. This indicates a significant reduction in vegetation cover within the highly affected zone. In the lowly affected area (10–15 km), a similar trend was observed with a -35.23% decrease in the area. However, the decrease was less pronounced compared to the highly affected area. Bareland in both highly and lowly affected areas, decreased from 2016 to 2022. However, the decrease was less in the lowly affected area

**Table 5. LULC changes in Kas locality between 2016 and 2022.**

| LULC class | 2016–2022 | | | |
|---|---|---|---|---|
| | Total area change | | Mean annual change rate | |
| | (km²) | (%) | (km²/year) | (%/year) |
| **Vegetation cover** | -10.20 | -35.33 | -1.70 | -5.89 |
| **Agricultural land** | 98.12 | 17.53 | 16.35 | 2.92 |
| **Bareland** | -66.85 | -18.98 | -11.14 | -3.16 |
| **Built-up** | 5.29 | 60.53 | 0.88 | 10.09 |
| **Sand** | -26.37 | -45.73 | -4.40 | -7.62 |

*Note. (-) = loss, (+) = Gain

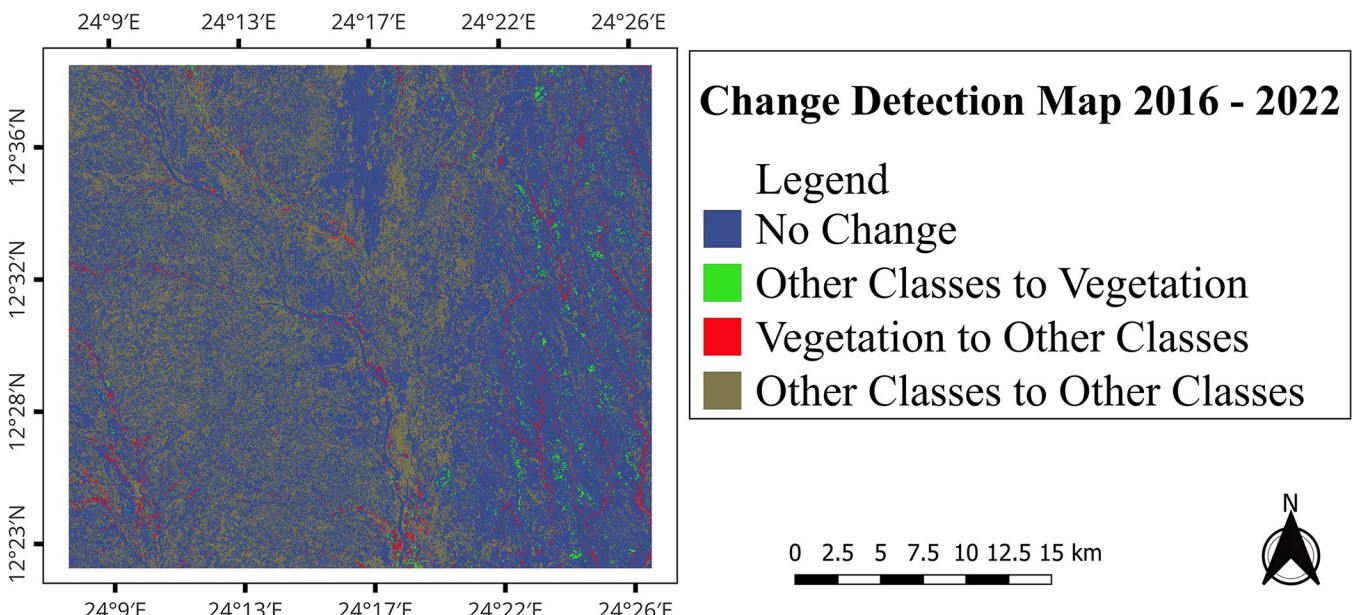

**Fig 9. Vegetation cover change detection map of 2016–2022.**

(-17.69%) compared to the highly affected area (-42.57%). Built-up in both highly and lowly affected areas experienced an increase, the percentage change in built-up areas was substantial, especially in the lowly affected area (102.56%), indicating rapid urbanization or infrastructure development in that zone. Agricultural land showed an increase in both highly and lowly affected areas, with a notable increase in the percentage area the change in agricultural land was more pronounced in the lowly affected area (17.70%) compared to the highly affected area (17.29%). There was a decrease in sand coverage in both highly and lowly affected areas, with a significant decrease in percentage area and actual area, the decrease was more substantial in the lowly affected area (-51.26%) compared to the highly affected area (-37.50%). The trends of the changes among different LULC classes suggest potential influences of human activities especially the IDPs, natural processes, and/or a combination of both in the study area.

**Table 6. Land cover areas and changes in the two buffer zones.**

| CLASS | Highly affected area (0-10km) | | | | | Lowly affected area (10-15km) | | | | |
|---|---|---|---|---|---|---|---|---|---|---|
| | 2016 | | 2022 | | 2016–2022 | 2016 | | 2022 | | 2016–2022 |
| | Area (%) | Area (km$^2$) | Area (%) | Area (km$^2$)$^2$ | Change (%) | Area (%) | Area (km$^2$) | Area (%) | Area (km$^2$) | Change (%) |
| **Vegetation Cover** | 2.32 | 10.01 | 1.33 | 5.80 | **-42. 57** | 3.47 | 19.87 | 2.25 | 12.87 | **-35.23** |
| **Bareland** | 37.81 | 165.51 | 31.12 | 135.32 | **-17.69** | 32.63 | 186.71 | 26.22 | 150.03 | **-19.65** |
| **Built-up** | 1.29 | 5.62 | 1.77 | 7.71 | **37.19** | 0.55 | 3.12 | 1.10 | 6.32 | **102.56** |
| **Agricultural Land** | 53.23 | 231.45 | 62.43 | 271.48 | **17.29** | 57.35 | 328.19 | 67.50 | 386.30 | **17.70** |
| **Sand** | 5.35 | 23.28 | 3.35 | 14.55 | **-37.50** | 6.00 | 34.39 | 2.93 | 16.76 | **-51.26** |
| **Total** | **100** | **434.86** | **100** | **434.8** | | **100** | **572.28** | **100** | **572.28** | |

Note: A decline is indicated by red highlights, and an increase is indicated by green highlights. Between 2016 and 2022, there was a noticeable decline in vegetation, bareland, and sand, and an increase in agricultural land and built-up classes in both highly and lowly affected zones.

**Table 7. LULC transition matrix (km2).**

|  |  |  |  | 2022 |  |  |  |
|---|---|---|---|---|---|---|---|
|  |  | **Vegetation Cover** | **Bareland** | **Built up** | **Agricultural Land** | **Sand** | **Total** |
|  | **Vegetation Cover** | 5.50 | 2.80 | 0.50 | 19.80 | 0.30 | 28.90 |
|  | **Bareland** | 1.10 | 159.30 | 3.40 | 179.20 | 9.20 | 352.20 |
| **2016** | **Built up** | 0.10 | 2.20 | 2.80 | 3.30 | 0.40 | 8.73 |
|  | **Agricultural Land** | 11.90 | 105.40 | 6.90 | 430.80 | 4.70 | 559.60 |
|  | **Sand** | 0.10 | 15.50 | 0.50 | 24.70 | 16.80 | 57.70 |
|  | **Total** | 18.70 | 285.30 | 14.10 | 657.80 | 31.40 | 1007.73 |

## Land cover change transition matrix

For a better understanding of the source and destination of the respective LULC changes, land cover change matrix for the two years of our study was computed (Table 7). LULC changes matrix analysis is important as it shows the direction of change and the LULC type that remains unchanged during the study period [58]. A large proportion of the vegetation cover (19.80 km$^2$) was converted to agricultural land while about 5.50 km$^2$ of vegetation remained unchanged between 2016 and 2022. Similarly, significant bareland areas (179.20 km$^2$) were converted to agricultural land. A minor proportion of the bareland was transformed to vegetation (1.10 km$^2$) and built-up area (3.40 km$^2$) over the study period Table 7.

## Discussion

The LULC classification output of this study were excellent as the overall accuracy were above the recommended threshold of 80% [56, 59]. Additionally, the Kappa statistics of the study showed a strong agreement of the classified image since accuracy was within the acceptable range to allow for further LULC changes detection assessment [58]. Five major LULC types (agricultural land, bareland, vegetation cover, sand and built-up) were classified for this study. Agricultural land and bareland were the dominant land cover types, with the former being by far the most dominant land cover type in the study area from 2016 to 2022. There were significant shifts in the areas of the different LULC classes with three classes showing a decline (vegetation cover, bareland, sand) while agricultural land and built-up areas increased. Vegetation was the second most reduced land cover type. The land cover class conversion matrix also revealed that vegetation to non-vegetation conversion increased substantially over the 6-year study period.

The vegetation cover loss in the Kas locality of Darfur can largely be attributed to resettlement of IDPs from the displaced populations in addition to other underlying factors. This was supported by the analysis results based on the 2 buffer zones (Fig 6, Table 6) which showed that vegetated areas within 10 km of IDPs settlements were highly affected relative to areas away from the settlements. The estimated populations of the IDPs in Kas locality has risen from between 35,000 to 40,000 in 2004 to about 77,843 by the year 2020 [34, 60]. Data from the United Nations Office for the Coordination of Humanitarian Affairs [27] further shows that the Darfur region of Sudan has been experiencing newly displaced populations annually from 2003 to 2014.

The resettlement of IDPs has subsequently increased the human and livestock populations in the Kas locality. The rise in number of IDPs subsequently resulted in extensive agricultural activities to meet their food demands hence the expansion of the agricultural land at the expense of vegetation cover and bareland as is evident in the transition matrix (Table 7). The increased demand for food production often leads to the conversion of the vegetated areas to

farms through clear felling of trees to increase the land for crop production. This negatively affects the vegetation cover area while contributing to the expansion of agricultural land [29]. This finding corroborates with that by [13], who reported the establishment of new agricultural areas on Azerbaijani territory, because of refugee migrations due to armed conflicts. A study by [61] further highlights the expansion of agricultural land, resettlement, and population growth as the key drivers of forest fragmentation in the Kaffa Biosphere Reserve of Ethiopia. Most IDPs are pastoralists who migrated with their livestock to the new settlement areas. This leads to increased livestock populations and overgrazing of the shrubs in the vegetated areas. Palatable tree branches are also cut down for fodder, contributing to deforestation. Similar findings were reported by [18] in the Afghan refugee camps in northern Pakistan.

Increase in human population due to conflict related displacements is frequently linked to increased deforestation rates due to overdependence and overutilization of the forest resources [19]. Increased human population in the neighbouring Zalingei IDP camps has previously been associated with considerable decrease in woody vegetation within the camp's vicinity [31]. A study in four hotspot regions around IDP camps in Darfur showed a correlation between decreased vegetation cover with logging of trees as well as removal of grass and shrubs [29]. The vegetation forms a key and readily accessible/available source of energy for the IDPs in the Kas locality in the form of firewood and charcoal [60]. Unregulated access, however, leads to overexploitation hence a reduction in vegetation cover. These findings conform with that by [14, 43, 62] who cite overexploitation of forest resources for firewood and charcoal production as a key driver of vegetation cover loss. The forests and shrubs further provide timber, building poles and tree branches which are used in the establishment of the temporary settlement structures by the IDPs (built-up areas) which has increased over time. This exerts additional pressure on the already dwindling vegetation cover in the study area. A study by [14, 19] in the Teknaf region of Bangladesh, showed that a mass influx of refugees coupled with a vast expansion of refugee camps led to large scale degradation of forestlands from overharvesting of poles and timber used in building the spontaneous settlements.

The major increase in the built-up areas can be linked to the the renewed civil conflict which erupted in 2014 emanating from fighting involving armed movements, government forces, and armed tribal militia which led to the displacement of more people hence more IDPs settlement. The built-up areas are likely to further increase with the recent conflicts between the Sudanese Armed Forces, and the paramilitary Rapid Support forces which started in April 2023, as the more people are likely to go back to the IDP camps. The area under sand notably far away from the IDPs camps and settlements substantially decreased, indicating that there has been a vegetation regeneration due to minimal disturbance.

Climate change and extreme negative climate variability have previously been reported in Sudan [22]. The variability of climatic conditions in the form of increased temperatures and variability of rainfall in the study area is also likely to play a role in the observed LULC changes [29]. There have been significant variations in the rainfall patterns and temperatures in the study area over the past three decades with an estimated 0.4˚C temperature rise per decade [29, 31]. Fluctuations in the seasonal precipitation patterns and temperature impacts the natural environment as they have the potential of creating drought-prone conditions [12, 31]. Although, climatic variations also influence the changes in the LULC, they play a minor role compared to the direct impacts of IDPs camp-related activities in this region [29, 63].

## Implications on environmental sustainability

Armed conflict and warfare primarily have a local impact on land systems, but they can also create tele couplings which can negatively affect the achievement of SDG6 (clean water and

sanitation), SDG13 (climate action), SDG15 (life on land) and SDG16 (peace justice and strong institutions) [64]. The armed conflicts in Sudan have resulted in the displacement of human and livestock populations. This coupled with the drought experienced in the last decade has also led to a massive displacement of the general population. These migrations have brought about overexploitation of forests, water resources, arable land, and pastures. The large numbers of IDPs living in refugee camps has further resulted in amplified groundwater pollution, air pollution and a rise in respiratory diseases caused by indoor cooking in the camp areas [60]. Additionally, there has been increased conflicts between herdsmen and farmers, worsened by the intervention of militias [65]. An effort by the government of Sudan to mitigate drought and climate change impacts, among others, has focused on the forestry sector with the establishment of new forest plantations [66]. Forests now are more important than ever before for they play a significant role in balancing the earth's carbon dioxide supply and exchange, thereby acting as a key link between the spheres of the earth [67]. Peacebuilding has the potential of boosting vegetation conservation; however, most conservation organizations' core mission in Sudan is not on conflict reduction. Partnerships between conservation and peace building organizations is therefore paramount for the reduction of future drastic land cover changes and the achievement of the above-mentioned SDGs [20].

## Conclusions

Using Sentinel-2 satellite data and GIS, this study sought to understand how the vegetation cover has changed in South Darfur State between 2016 and 2022 largely due to IDPs settlement among other factors, brought about by renewed armed conflicts in year 2014. Study findings show a considerable negative impact of the IDPs on vegetation cover, as seen by the negative change trend. Since the IDPs depend on the forest, shrubs, and scattered trees for their livelihoods, vegetation cover area has experienced a significant reduction due to over exploitation, which affects the natural regeneration process. Other natural and human driven activities like climate change, traditional rain-fed agricultural practices and grazing have also led to a reduction in vegetation cover in the Kas locality.

These findings underscore the urgent need for targeted interventions and policy responses to address the environmental consequences of conflict-driven displacement. Whereas the consequences armed conflict is typically felt immediately, they have the potential to create long lasting effects on land cover and the achievement of SDGs. Peace efforts can offer long lasting solutions as the IDPs will return to their homes thereby reducing the pressure on the vegetation cover. Implementation of sustainable agricultural management practices and regulated utilization of natural ecosystems like forests and shrubs can help curb vegetation cover loss in the Kas locality. The study offers pertinent data that can help decision-makers plan restoration of the degraded areas and prevent further environmental degradation to promote long-term environmental sustainability. Recommendations for future studies include a detailed social survey and key informant interviews in the study area, when peace returns, to establish the other underlying factors contributing to LULC changes.

## Supporting information

**S1 File. Stack bands of 2022 S2 image.**
(7Z)

**S2 File. Training and testing samples.**
(ZIP)

**S3 File. Stack bands of 2016 S2 image.**
(ZIP)

## Acknowledgments

Much gratitude goes to the Sudanese Ministry of Higher Education and Scientific Research for their academic support, as well as the Tempus Public Foundation and the University of Sopron for providing a PhD scholarship to the first author.

## Author Contributions

**Conceptualization:** Abdalrahman Ahmed.

**Formal analysis:** Abdalrahman Ahmed.

**Methodology:** Abdalrahman Ahmed, Brian Rotich.

**Software:** Abdalrahman Ahmed.

**Supervision:** Kornel Czimber.

**Visualization:** Abdalrahman Ahmed, Brian Rotich.

**Writing – original draft:** Abdalrahman Ahmed, Brian Rotich.

**Writing – review & editing:** Brian Rotich, Kornel Czimber.

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
