## [Decision Letter · Decision Letter 0]

9 Feb 2024

PONE-D-24-01879Assessment of the environmental impacts of conflict-driven Internally Displaced Persons: A sentinel-2 satellite based analysis of vegetation cover changes in the Kas locality, Darfur, SudanPLOS ONE

Dear Dr. Ahmed,

Thank you for submitting your manuscript to PLOS ONE. After careful consideration, we feel that it has merit but does not fully meet PLOS ONE’s publication criteria as it currently stands. Therefore, we invite you to submit a revised version of the manuscript that addresses the points raised during the review process.

We look forward to receiving your revised manuscript.

Kind regards,

Bijeesh Kozhikkodan Veettil

Academic Editor

PLOS ONE

Journal Requirements:

2. We note that your Data Availability Statement is currently as follows: "All relevant data are within the manuscript and its Supporting Information files."

3. We note that Figures 1, 3, 6, 7  in your submission contain map images which may be copyrighted. All PLOS content is published under the Creative Commons Attribution License (CC BY 4.0), which means that the manuscript, images, and Supporting Information files will be freely available online, and any third party is permitted to access, download, copy, distribute, and use these materials in any way, even commercially, with proper attribution. For these reasons, we cannot publish previously copyrighted maps or satellite images created using proprietary data, such as Google software (Google Maps, Street View, and Earth). For more information, see our copyright guidelines: http://journals.plos.org/plosone/s/licenses-and-copyright.

     1. You may seek permission from the original copyright holder of Figures 1, 3, 6, 7 to publish the content specifically under the CC BY 4.0 license. 

Reviewers' comments:

Reviewer's Responses to Questions

**Comments to the Author**

1. Is the manuscript technically sound, and do the data support the conclusions?

Reviewer #1: No

Reviewer #2: Partly

2. Has the statistical analysis been performed appropriately and rigorously? 

Reviewer #1: No

Reviewer #2: Yes

3. Have the authors made all data underlying the findings in their manuscript fully available?

Reviewer #1: Yes

Reviewer #2: Yes

4. Is the manuscript presented in an intelligible fashion and written in standard English?

Reviewer #1: Yes

Reviewer #2: Yes

5. Review Comments to the Author

Reviewer #1: 1. The manuscript title suggests an assessment of the environmental impacts of conflict-driven internally displaced ---

persons (IDPs). It is important to note that while the study may identify changes in vegetation cover, establishing a

direct causal link between these changes and IDPs may be challenging. Other factors may also contribute to

vegetation cover changes.

2. The choice of using only Random Forest (RF) classifier used in the research article can influence the accuracy and

performance of land cover classification. Different algorithms have varying strengths and weaknesses, and their

suitability depends on the specific study area and land cover types. It is important to consider and compare other

algorithm(s) and evaluate their performance

3. The novelty of this paper is lacking. It is important to emphasize the distinctions between the current study and

previous studies.

4. Include recommendation for future studies.

Reviewer #2: In this paper, land use/land cover changes in the Kas locality of Darfur, Sudan were detected during the years 2016 to 2022 using a cross-tabulation approach and S2 images. Then, these changes, solely descriptively, were attributed to some driving factors such as resettlement of Internally Displaced Persons and Climate change. However, this differs from the work stated in the title of the paper, i.e. Assessment of the environmental impacts of conflict driven Internally Displaced Persons: A sentinel-2 satellite based analysis of vegetation cover changes in the Kas locality, Darfur, Sudan. In order to perform the work stated in the paper title, it is required to separate the impacts of conflict driven Internally Displaced Persons on land cover changes from those of other affecting factors, and then analyze them. Therefore, it is suggested to change the title, According to the work that was actually done in this research. A title like “Detecting vegetation cover changes in the Kas locality, Darfur, Sudan using S2 images and investigating their affecting factors” can be more appropriate.

There are some suggestions, questions, and minor corrections in the attached file that I respectfully ask the authors to address them.

Best regards.

6. PLOS authors have the option to publish the peer review history of their article (what does this mean?). If published, this will include your full peer review and any attached files.

Reviewer #1: No

Reviewer #2: No

---

## [Author Response · Author response to Decision Letter 0]

22 Mar 2024

RESPONSE TO REVIEWERS

Dear reviewers,

Thank you very much for your comments and suggestions on our paper titled “Assessment of the environmental impacts of conflict-driven Internally Displaced Persons: A sentinel-2 satellite-based analysis of vegetation cover changes in the Kas locality, Darfur, Sudan”. The comments are very insightful and helpful in improving the quality of our paper and provide important guidance to our research. We have carefully considered and paid special attention to every comment and have responded to each one of them as detailed below. The response to reviewers’ comments is in green color, while the corresponding changes to the text are highlighted in track changes in the revised version of the manuscript. We sincerely hope that the revisions will be satisfactory for your approval.

Kind regards

Corresponding author

Reviewer #1

 1. The manuscript title suggests an assessment of the environmental impacts of conflict-driven internally displaced persons (IDPs). It is important to note that while the study may identify changes in vegetation cover, establishing a direct causal link between these changes and IDPs may be challenging. Other factors may also contribute to vegetation cover changes.

Thank you for your critical observation. We have expanded the study area to demonstrate and gain a deeper comprehension of the IDP settlement impacts on changes in vegetation cover (Figure 4, Table 6). To do this, we employed the SAGA tool in QGIS version 3.28.2 After applying a majority filter with a 20-radius, we isolated the built-up class using a raster calculator. Lastly, we created a buffer of 10 km from the settlement area; we classify this area as IDPs and climate change affected region (highly affected) and the remaining area as climate change affected region (low affected) (Figure 4)

Despite IDPs being the main contributor to the changes in vegetation cover in the study area, there may be other underlying factors that also contribute to vegetation changes. We have acknowledged the contribution of other factors to vegetation cover changes in the discussion section as follows “The vegetation cover loss in the Kas locality of Darfur can be largely attributed to resettlement of IDPs from the displaced populations, in addition to other underlying factors” … “The variability of climatic conditions in the form of increased temperatures and variability of rainfall in the study area is also likely to play a role in the observed LULC changes [1] There have been significant variations in the rainfall patterns and temperatures in the study area over the past three decades with an estimated 0.4°C temperature rise per decade [1,2] Fluctuations in the seasonal precipitation patterns and temperature impacts the natural environment as they have the potential of creating drought-prone conditions [2,3] “ and conclusion section as follows: “Using Sentinel-2 satellite data and GIS, this study sought to understand how the vegetation cover has changed in South Darfur State between 2016 and 2022 largely because of IDP settlement, among other factors, brought about by renewed armed conflicts in the year 2014”.

2. The choice of using only Random Forest (RF) classifier used in the research article can influence the accuracy and performance of land cover classification. Different algorithms have varying strengths and weaknesses, and their suitability depends on the specific study area and land cover types. It is important to consider and compare other algorithm(s) and evaluate their performance.

Thank you for your insightful recommendation. In addition to the Random Forest (RF) classifier, we also considered and compared the Support Vector Machine (SVM), and K-Nearest Neighbor (KNN) algorithms. The RF had a higher overall accuracy and kappa hat for both study years compared to the other classifiers (Table 3) hence our decision to adopt the RF classifier for this study. Additionally, the random forest classifier is a powerful machine learning technique since it applies feature significance properties and averages several predictions.

Table 3. Overall accuracy (%) and kappa coefficient values of Three different Machine Learning (ML) algorithms used to classify Sentinel 2 satellite imagery.

3. The novelty of this paper is lacking. It is important to emphasize the distinctions between the current study and previous studies.

Thank you for this vital comment. We have emphasized the novelty of the study in the last paragraph of the introduction section as follows: “Previous vegetation cover change research in Darfur [4–6] was done before 2016. All those studies were conducted in Central, Western, and Nouth Darfur. Our study is the first after the launch of Sentinel-2 satellite data for public use. This study therefore aims to provide the latest essential post 2014 spatial data on LULC changes in the Kas locality of Darfur, which is lacking owing to the ongoing conflicts. The results will further enhance the understanding of the repercussions of the 2014 conflict-driven IDPs on vegetation cover and overall environmental sustainability in the Kas locality of Darfur, Sudan.”

4. Include recommendation for future studies.

We are grateful for your suggestion and have revised the conclusion section to Include recommendation for future studies as follows… “Recommendation for future studies include a detailed social survey and Key informant interviews in the study area, when peace returns, to establish the other factors contributing to LULC changes in the Kas locality”. 

Reviewer #2

In this paper, land use/land cover changes in the Kas locality of Darfur, Sudan were detected during the years 2016 to 2022 using a cross-tabulation approach and S2 images. Then, these changes, solely descriptively, were attributed to some driving factors such as resettlement of Internally Displaced Persons and Climate change. However, this differs from the work stated in the title of the paper, i.e. Assessment of the environmental impacts of conflict driven Internally Displaced Persons: A sentinel-2 satellite based analysis of vegetation cover changes in the Kas locality, Darfur, Sudan. In order to perform the work stated in the paper title, it is required to separate the impacts of conflict driven Internally Displaced Persons on land cover changes from those of other affecting factors, and then analyze them. Therefore, it is suggested to change the title, according to the work that was actually done in this research. A title like “Detecting vegetation cover changes in the Kas locality, Darfur, Sudan using S2 images and investigating their affecting factors” can be more appropriate.

Many thanks for this suggestion. We have carried out further analysis on the images (Figure 4, Table 6) to separate the impacts of conflict driven Internally Displaced Persons on land cover changes from those of other affecting factors as follows “After the LULC was successfully classified, we subsequently used the Majority/minority filter in QGIS's SAGA tool to isolate the built-up class using a 20 radius. A raster buffer was made 10 km from the center of the IDPs settlements (Fig 4, Table 6) for comparative analysis of the vegetation cover in the immediate 10 km and next 5km buffer zones and to establish which of the two zones was highly affected. This analysis was conducted to help us better understand the changes in vegetation cover caused by the IDPs as opposed to other underlying drivers”.

Suggestions, questions, and minor corrections in the manuscript. 

This feature of S2 has been neglected in this study! The use of a time-series of S2 images could lead to a more detailed investigation of the changes in the study area.

Thank you for the observation. We have revised this part of the citation.

Produce

“Produces” changed to “produce”

About 322000 people or more than 320000 people.

About 321, 929 Changed to about 322,000 people.

regions.

Full stop added

and its average annual rainfall

“its” added to the sentence 

S2 is not a sensor. Its sensor is the multi-spectral instrument (MSI).

Table 1 corrected and S2 replaced with multi-spectral instrument (MSI)

Which level of processing was used for the level-1 data (level-1A, level-1B, or level-1C)? You could use S2 level-2A data which are presented as bottom of atmosphere reflectance, and do not require atmospheric correction.

Thank you for this suggestion. For the previous images, Sentinel 2 level-2A data were not available on 08-12-2016 when the images were acquired. Based on the recommendation, we have modified the image acquisition dates to 12-02-2016 and 20-02-2022 to utilize S2 level-2A data (Table 1). 

Red: B4, Green: B3, and Blue: B2. However, NIR pseudo color composite is suggested (i.e. Red: B8, Green: B4, and Blue: B3). 

Thank you for the suggestion. the NIR pseudo color (red: B8, Green: B4, and Blue: B3) have been utilized in addition to Red: B4, Green: B3, and Blue: B2.

How did you tune the RF model? More precisely, how was the RF hyper-parameters such as the number and type of trees, and the number of features tuned?. 

Thank you for the inquiry. We used the default parameters setting of the Dzetsaka plugin in QGIS, which by default has a fixed number of 100 trees set. We have subsequently updated the statement to “A total of three classifiers—random forest (RF), support vector machine (SVM), and k-nearest neighbor (KNN)—were used to classify the Sentinel-2 images. The RF classifier is an ensemble method using decision trees as classifiers. For it to function, a large number of decision trees are feature-aggregated (bagging) using the bootstrap of the training sample. The final output is determined by the majority of the trees. [7,8]. The Dzetsaka classification tool in QGIS was utilized in this study to apply RF classification to the Sentinel-2A images. The random forest is a powerful learning technique since it applies feature significance properties and averages several predictions. A fixed number of 100 trees is set by default, and this has shown to be an appropriate size to avoid overfitting. [9–11] .During the construction each tree is split at every internal node using the square root of the number of features (n):

max_features = √((n_features)) (1)

From the training data (n_features), which consists of the pixel values defined by the ROIs (regions of interest), the features are determined and chosen at random. Thus, a suitable size for the training data set should be ensured. There is no predetermined size for this; it relies on the specific attributes of each data set and the intended class number. [12–14].

The KNN is a nonparametric memory-based supervised machine learning classifier. Following the calculation of the number of neighbors for which k is an integer value [15] KNN is used to solve both classification and regression problems [16]. This parameter (number of neighbors) in the Dzetsaka plugin is selected using a cross-validation technique to optimize the quality of output. [17]. KNN classification was applied to the study area using QGIS's dzetsaka classification tool.

SVM, is a linear model for classification and regression problems that is mainly based on kernels, it was developed by Cortes and Vapnik [18]. The Gaussian kernel known as the radial basis function is the one used in Dzetsaka and provides high quality results for classifying tree species. [17]. SVMs have been utilized in this study as it widely used in remote sensing [19]. A mathematical formulation of the SVM can be found in Scikit learn [13,20]”

What features were considered as inputs of the RF classifier? Did they only include the S2 bands (B2, B3, b4, AND b8)? It is suggested to use some vegetation indices (such as NDVI or SAVI), built-up indices (such as NDBI), texture analysis components.

Many thanks for the suggestion. We have included the NDVI as an input of the RF classifier and we also added the NDVI map in the manuscript (Fig 3). 

Use the same term for the same concept. Both error matrix and confusion matrix are correct. But be consistent in using terms. Confusion matrix is a more common term. 

Thank you for the suggestion. The confusion matrix has been adopted as suggested and used consistently.

This is not true. It is possible for Kappa to be negative. For a completely random classification, the kappa value is expected to be close to zero, and for worse classification (than a random classification), kappa becomes negative.

Table 4?

We appreciate your correction; a negative Kappa statistic can indeed occur. We have revised the statement to read, "The Kappa coefficient can be negative and range from 0 (showing no agreement) to 1 (perfect agreement). The kappa value should be around zero for a fully random classification, and negative for a worse classification (than a random classification) [21,22]”

How do you explain the time difference between these statistics (2003-2014) and the satellite images used for LULC change detection (2016-2022)?

We appreciate this significant observation. The statistics were meant to show the general trend of displaced persons in Darfur since the year 2003. The data were available up to the year 2014 when the ruling government halted the operation of most humanitarian organizations and NGOs in Darfur hence the lack of continuity of the official statistics on IDPs. We have therefore removed the table to avoid confusion. 

What are these numbers? Are these the references used? Check it out.

The numbers represent the SDGs. We have revised them for clarity as follows: “…negatively affect the achievement of SDG6 (clean water and sanitation), SDG13 (climate action), SDG15 (life on land) and SDG16 peace justice and strong institutions)”.

Is there any evidence to demonstrate that the detected vegetation cover changes were exclusively due to IDPs settlement?

Thank you for your constructive comment. IDP settlement is the leading contributor to vegetation cover changes but not the exclusive cause of the changes (See details on the response to comment 1 above).

Additional references 

[1] Kranz O, Sachs A, Lang S. Assessment of environmental changes induced by internally displaced person (IDP) camps in the Darfur region, Sudan, based on multitemporal MODIS data. Int J Remote Sens 2015;36:190–210. https://doi.org/10.1080/01431161.2014.999386.

[2] Spröhnle K, Kranz O, Schoepfer E, Moeller M, Voigt S. Earth observation-based multi-scale impact assessment of internally displaced person (IDP) camps on wood resources in Zalingei, Darfur. Geocarto Int 2016;31:575–95. https://doi.org/10.1080/10106049.2015.1062053.

[3] Mohamed MA, Anders J, Schneider C. Monitoring of changes in land use/land cover in Syria from 2010 to 2018 using multitemporal landsat imagery and GIS. Land (Basel) 2020;9. https://doi.org/10.3390/land9070226.

[4] Spröhnle K, Kranz O, Schoepfer E, Moeller M, Voigt S. Earth observation-based multi-scale impact assessment of internally displaced person (IDP) camps on wood resources in Zalingei, Darfur. Geocarto Int 2016;31:575–95. https://doi.org/10.1080/10106049.2015.1062053.

[5] Kranz O, Zeug G, Tiede D, Clandillon S, Bruckert D, Kemper T, et al. Monitoring Refugee/IDP camps to Support International Relief Action. Geoinformation for Disaster and Risk Management - Examples and Best Practices 2010:51–6.

[6] Kranz O, Sachs A, Lang S. Assessment of environmental changes induced by internally displaced person (IDP) camps in the Darfur region, Sudan, based on multitemporal MODIS data. Int J Remote Sens 2015;36:190–210. https://doi.org/10.1080/01431161.2014.999386.

[7] Ghosh A, Sharma R, Joshi PK. Random forest classification of urban landscape using Landsat archive and ancillary data: Combining seasonal maps with decision level fusion. Applied Geography 2014;48:31–41. https://doi.org/10.1016/j.apgeog.2014.01.003.

[8] Maxwell AE, Warner TA, Fang F. Implementation of machine-learning classification in remote sensing: An applied review. Int J Remote Sens 2018;39:2784–817. https://doi.org/10.1080/01431161.2018.1433343.

[9] Breiman L. Random Forests. vol. 45. 2001.

[10] Sudhakar V, Venkata Sudhakar C, Reddy GU. Limestone mining area mapping and assessment at the Cement Industrial area using Spectral Index and

---

## [Decision Letter · Decision Letter 1]

8 Apr 2024

PONE-D-24-01879R1Assessment of the environmental impacts of conflict-driven Internally Displaced Persons: A sentinel-2 satellite based analysis of vegetation cover changes in the Kas locality, Darfur, SudanPLOS ONE

Dear Dr. Ahmed,

Thank you for submitting your manuscript to PLOS ONE. After careful consideration, we feel that it has merit but does not fully meet PLOS ONE’s publication criteria as it currently stands. Therefore, we invite you to submit a revised version of the manuscript that addresses the points raised during the review process.

We look forward to receiving your revised manuscript.

Kind regards,

Bijeesh Kozhikkodan Veettil

Academic Editor

PLOS ONE

Journal Requirements:

Reviewers' comments:

Reviewer's Responses to Questions

**Comments to the Author**

1. If the authors have adequately addressed your comments raised in a previous round of review and you feel that this manuscript is now acceptable for publication, you may indicate that here to bypass the “Comments to the Author” section, enter your conflict of interest statement in the “Confidential to Editor” section, and submit your "Accept" recommendation.

Reviewer #1: All comments have been addressed

2. Is the manuscript technically sound, and do the data support the conclusions?

Reviewer #1: Yes

3. Has the statistical analysis been performed appropriately and rigorously? 

Reviewer #1: Yes

4. Have the authors made all data underlying the findings in their manuscript fully available?

Reviewer #1: Yes

5. Is the manuscript presented in an intelligible fashion and written in standard English?

Reviewer #1: (No Response)

6. Review Comments to the Author

Reviewer #1: Change the phrase in the title

"A sentinel-2 satellite based analysis of vegetation cover changes"

to

"A sentinel-2 satellite based analysis of land use/cover changes"

The change from "vegetation cover changes" to "land use/cover changes" broadens the scope of the research, allowing for a more comprehensive analysis of the environmental impacts of conflict-driven displacement.

7. PLOS authors have the option to publish the peer review history of their article (what does this mean?). If published, this will include your full peer review and any attached files.

Reviewer #1: No

---

## [Author Response · Author response to Decision Letter 1]

11 Apr 2024

RESPONSE TO REVIEWERS

Dear reviewer,

We greatly appreciate your input on our paper “Assessment of the environmental impacts of conflict-driven Internally Displaced Persons: A sentinel-2 satellite based analysis of vegetation cover changes in the Kas locality, Darfur, Sudan”. After thorough consideration, we have incorporated the suggested change, which has expanded the scope of our study as outlined below. The response to reviewer’ has been addressed in green color, while the corresponding changes to the text are highlighted in track changes in the revised version of the manuscript.

Kind regards

Corresponding author

Reviewer #1

Reviewer #1: Change the phrase in the title

"A sentinel-2 satellite based analysis of vegetation cover changes"

to

"A sentinel-2 satellite based analysis of land use/cover changes"

Thank you for the suggestion. We have changed the phrase to “A sentinel-2 satellite based analysis of land use/cover changes”. The new title now reads “Assessment of the environmental impacts of conflict-driven Internally Displaced Persons: A sentinel-2 satellite based analysis of land use/cover changes in the Kas locality, Darfur, Sudan”.

---

## [Editor Report · Decision Letter 2]

15 Apr 2024

Assessment of the environmental impacts of conflict-driven Internally Displaced Persons: A sentinel-2 satellite based analysis of land use/cover changes in the Kas locality, Darfur, Sudan

PONE-D-24-01879R2

Dear Dr. Ahmed,

We’re pleased to inform you that your manuscript has been judged scientifically suitable for publication and will be formally accepted for publication once it meets all outstanding technical requirements.

Kind regards,

Bijeesh Kozhikkodan Veettil

Academic Editor

PLOS ONE
---

## [Editor Report · Acceptance letter]

7 May 2024

PONE-D-24-01879R2 

PLOS ONE

Dear Dr. Ahmed, 

I'm pleased to inform you that your manuscript has been deemed suitable for publication in PLOS ONE. Congratulations! Your manuscript is now being handed over to our production team.

Kind regards, 

on behalf of

Dr. Bijeesh Kozhikkodan Veettil 

Academic Editor

PLOS ONE